# White Adipose Tissue Heterogeneity in the Single-Cell Era: From Mice and Humans to Cattle

**DOI:** 10.3390/biology12101289

**Published:** 2023-09-27

**Authors:** Hunter Ford, Qianglin Liu, Xing Fu, Clarissa Strieder-Barboza

**Affiliations:** 1Department of Veterinary Sciences, Davis College of Agricultural Sciences and Natural Resources, Texas Tech University, Lubbock, TX 79409, USA; hunter.ford@ttu.edu; 2School of Animal Sciences, Agricultural Center, Louisiana State University, Baton Rouge, LA 70803, USA; qliu19@lsu.edu (Q.L.); xfu1@agcenter.lsu.edu (X.F.); 3School of Veterinary Medicine, Texas Tech University, Amarillo, TX 79106, USA

**Keywords:** adipose, single-cell, sequencing, cattle, livestock, transcriptional diversity

## Abstract

**Simple Summary:**

Progress in adipose research has shifted our understanding of adipose tissue from being a homogenous, quiescent depot for energy storage to a highly dynamic organ with wide-ranging roles in whole-body health and metabolism with distinct, depot-specific functional differences. Through advances in genomic technologies, particularly the application of single-cell sequencing techniques, the vast cellular heterogeneity of white adipose tissue depots has been elucidated, providing insight into unique cell populations that contribute to functional differences. Furthermore, the utilization of these techniques has advanced our understanding of the pathogenesis of metabolic diseases such as obesity and type 2 diabetes. Recent studies in livestock highlight the potential of these approaches to improve animal health and productivity, although research in this field is still in its early stages.

**Abstract:**

Adipose tissue is a major modulator of metabolic function by regulating energy storage and by acting as an endocrine organ through the secretion of adipokines. With the advantage of next-generation sequencing-based single-cell technologies, adipose tissue has been studied at single-cell resolution, thus providing unbiased insight into its molecular composition. Recent single-cell RNA sequencing studies in human and mouse models have dissected the transcriptional cellular heterogeneity of subcutaneous (SAT), visceral (VAT), and intramuscular (IMAT) white adipose tissue depots and revealed unique populations of adipose tissue progenitor cells, mature adipocytes, immune cell, vascular cells, and mesothelial cells that play direct roles on adipose tissue function and the development of metabolic disorders. In livestock species, especially in bovine, significant gaps of knowledge remain in elucidating the roles of adipose tissue cell types and depots on driving the pathogenesis of metabolic disorders and the distinct fat deposition in VAT, SAT, and IMAT in meat animals. This review summarizes the current knowledge on the transcriptional and functional cellular diversity of white adipose tissue revealed by single-cell approaches and highlights the depot-specific function of adipose tissue in different mammalian species, with a particular focus on recent findings and future implications in cattle.

## 1. Introduction

Advances in research have revealed white adipose tissue as a complex organ that modulates metabolism and health via endocrine and other signaling mechanisms [1]. As such, considerable efforts have been made to further our understanding of this tissue, particularly as it relates to metabolic dysfunction and disease. These efforts have primarily focused on adipose-related disorders in humans, such as obesity and type 2 diabetes, with limited studies in cattle and other livestock.

Investigations in recent decades have demonstrated that not all white adipose tissue depots are the same, with distinct functional differences between subcutaneous, visceral, and intramuscular sites [2]. Furthermore, these depot-specific functional differences are driven by significant cellular heterogeneity [3], resulting in distinct metabolic, immune, and inflammatory profiles. Early studies using antibodies to identify and isolate unique cell populations in adipose tissue depots were critical for beginning to understand the cellular diversity within these tissues [4]; however, the advent of single-cell/nucleus RNA sequencing (sc/snRNAseq) techniques has revolutionized the field and opened up new avenues of exploration. As evidenced by research endeavors in cancer biology [5], the application of sc/snRNAseq technologies in the field of adipose tissue biology has great potential to further enhance our understanding of how adipose tissue contributes to metabolic function and the pathogenesis of disease in both humans and livestock.

In this comprehensive review, we briefly discuss the functions of white adipose tissue, highlighting distinct depot-specific differences and provide a review of sc/snRNAseq studies that address the cell type composition of subcutaneous (SAT), visceral (VAT), and intramuscular (IMAT) adipose tissue depots, with particular attention on adipose stem/progenitor cells (ASPCs) and adipocytes. We highlight recent sc/snRNAseq studies conducted on cattle and other agricultural species, shedding light on the latest advancements in our understanding of this crucial tissue and future implications for dairy and beef research.

## 2. It Is Not Just Fat: Adipose Tissue Composition and Function

### 2.1. Adipose Tissue and Its Main Cellular and Non-Cellular Components

The two main types of white adipose tissue depots in animals are SAT and VAT. As the name indicates, SAT is located throughout the body underneath the skin and functions as the primary storage site of excess lipids during periods of positive energy balance. Visceral adipose tissue is located around the internal organs and in the abdominal cavity and in some species functions as a secondary lipid storage location when the storage capacity of SAT is exceeded [6]. In humans, SAT makes up ~80% of total body fat, whereas VAT makes up ~10–20% of total body fat in men and ~5–8% of total body fat in women, with the % of VAT increasing with age in both sexes [7]. When nutrient availability exceeds the adipose tissue’s ability to accumulate and incorporate lipids, fat is deposited in other locations, such as the in the liver, heart, or muscle, often with negative effects on tissue functionality [8,9]. Conversely, in beef cattle, the deposition of IMAT (also known as marbling) is desirable [10] as it improves meat quality and consumer acceptability [11].

Although key differences exist among white adipose tissue depots in the types of cells they contain, endocrine function, and metabolic activity, the general structure of white adipose tissue depots is similar. The primary cell type in white adipose tissue depots is unilocular adipocytes. These cells uptake free fatty acids (FFAs) from circulation and store them intracellularly as triglycerides (TG) in a single, large lipid droplet that occupies most of the cell volume. Smaller adipocytes have greater insulin sensitivity and FFAs uptake activity than larger adipocytes, whereas larger adipocytes are more insulin-resistant and hyperlipolytic [12] and secrete more pro-inflammatory cytokines [13] than smaller adipocytes. In dairy cows, omental VAT has smaller adipocytes, but an increased adipocyte number per gram of tissue compared with abdominal SAT [14]. Differences in adipocyte lipogenesis seem to be mediated by the higher expression of adipogenic and lipogenic genes in subcutaneous adipocytes compared to visceral omental cells [15]. Functionally, larger SAT adipocytes have demonstrated increased basal and isoproterenol-stimulated lipolysis than omental fat, but are less responsive to insulin inhibition of lipolysis in dairy cows [14]. In late-gestating dairy cows, similar adipocyte area and diameter has been reported between subcutaneous, omental, intrapelvic, and perirenal adipocytes, but marked differences were observed when compared with mesenteric cells, which are significantly smaller [16].

The stromal vascular fraction (SVF) of white adipose tissue contains populations of multipotent stem cells, fibroblasts, preadipocytes, vascular cells, and immune cells [17]. Stem cells present in the SVF fulfill a wide range of functions including the expansion of adipose tissue via hyperplasia and angiogenesis, as well as the capacity to differentiate into a wide variety of cell types [18]. Recently, transcriptionally and functionally distinct populations of adipose stem/progenitor cells (ASPC) were identified, isolated, and characterized in human and animal models, and will be further discussed below [19,20,21]. Before that, it was generally thought not only ASPCs but also adipocytes were homogenous populations of cells playing similar roles in the adipose tissue. Macrophages, lymphocytes, neutrophils, and mast cells are also present in adipose tissue, typically recruited by the secretion of chemokines and inflammatory mediators by adipocytes. The infiltration of immune cells into adipose tissue is implicated in the chronic inflammation experienced by individuals with obesity as well as insulin resistance [22]. In dairy cattle, increased macrophage infiltration into SAT has been associated with increased body condition score loss in the postpartum period and increased local expression of the chemotactic cytokines, such as C-C motif chemokine ligand 22 (*CCL22*), osteopontin (*SPP1*), and the receptor for SPP1, cluster of differentiation 44 (*CD44*) [23]. Additionally, macrophage infiltration in adipose tissue is increased in postpartum dairy cows with displaced abomasum, particularly in the omental depot compared with SAT [24]. Increased lipid mobilization around parturition time in dairy cows seems to be the triggering factor for inducing higher immune cell infiltration into adipose tissue and triggers a deep adipose tissue remodeling process involving changes in inflammatory responses, vascularization, cell proliferation, and extracellular matrix (ECM) structure, as previously reviewed [25,26].

The extracellular matrix provides mechanical support to adipose tissue and is composed mostly of collagen, non-collagen proteins, and proteoglycans produced by adipocytes as well as other cell types in the stromal vascular fraction [27]. Crosstalk between the ECM and cells within the adipose tissue is critical for proper adipose tissue function and has been implicated in numerous cellular processes including adipocyte differentiation and inflammation as well as in the pathogenesis of obesity and metabolic dysfunction in human and mouse models [27,28,29]. In cattle, the role of adipose tissue ECM and how it may alter adipocyte and other cells function in healthy and disease states remain a gap in knowledge.

Blood supply to adipose tissue is supported by a dense network of capillaries [30]; however, under conditions of metabolic dysfunction, such as insulin resistance or obesity, capillary networks decrease and are replaced by networks of larger blood vessels [31]. There are limited studies evaluating angiogenesis in adipose tissue from livestock; however, Yamada et al. [32] observed depot-specific differences in vascular endothelial growth factor (*VEGF*) and fibroblast growth factor (*FGF*) expression, with visceral and intermuscular depots expressing higher levels of these pro-angiogenic factors than subcutaneous tissue. Our recent work has identified vascular and lymphatic endothelial cell populations in both SAT and VAT of dairy cows. However, a higher frequency of these cells were observed in SAT, thus suggesting an increased angiogenic capacity of this depot compared to VAT [33].

Innervation of white adipose tissue depots via the sympathetic nervous system (SNS) provides an important link between cellular and nervous regulation of lipid metabolism [34,35,36]. In general, neuronal control of lipid mobilization occurs via activation of pro-lipolytic β-adrenergic receptors (βARs) or anti-lipolytic α-adrenergic receptors (αARs) by catecholamines [37], with increased responsiveness to catecholamines contributing to enhanced lipolysis in adipose tissue of dairy cows in the early postpartum [38]. However, the dynamics of how white adipose tissue is innervated, and its effects remain relatively understudied, particularly in ruminants.

### 2.2. Major Depot-Specific Functional Differences

White adipose tissue is found in distinct anatomical locations and typically divided into SAT, VAT, IMAT, and intermuscular fat [39]. While intermuscular fat accumulates between skeletal muscles and is the most abundant adipose tissue depot in beef steers [40], IMAT deposits among skeletal muscle fibers [39]. As reviewed by Hausman et al. [39], intermuscular fat shares many transcriptional similarities with VAT in both humans and pigs, characterized by increased expression of immune and inflammatory markers. Intramuscular adipose tissue is negligible in most species but prevalent in certain meat animal species, particularly beef cattle and sheep breeds developed for lamb meat production [41]. In beef cattle, unlike the SAT and VAT depots which are generally considered waste, IMAT provides lubricative and flavory effects to meat and is a characteristic of high-quality beef [42]. In contrast, IMAT deposition in humans is generally associated with degenerative myopathies, such as muscular dystrophies and sarcopenia [43,44,45,46]. It is worth noting that despite some histological similarities, including fatty infiltration into perimysium and endomysium, unlike IMAT associated with human myopathy, massive muscular degeneration and inflammation are often not observed in bovine IMAT, suggesting the presence of distinct underlying mechanisms [47,48,49].

In general, SAT depots function as the primary site of lipid storage, typically exhibiting higher adipogenic capacity than other adipose tissue depots [6]. In dairy heifers, adipogenesis (assessed by ASPC number) is lower in omental and mesenteric VAT, as well as in tailhead, withers, and sternum SAT compared with retroperitoneal VAT [50]. Increased number of ASPCs enhances the adipose tissue adipogenic capacity and the potential to store lipids, a desirable state for reducing the excess circulating FFA in periparturient cows [26,51,52]. In vitro, we previously revealed that SAT preadipocytes from tailhead accumulate more lipids and have higher expression of the adipogenic markers *ADIPOQ*, *CEBPA*, *CEBPB*, and *PPARG* compared with omental VAT in dairy cows [15]. In contrast, both VAT and IMAT depots have higher rates of lipolysis than SAT depots and contribute to insulin resistance and metabolic dysfunction in humans [53]. Furthermore, VAT and IMAT depots exhibit distinct pro-inflammatory profiles in humans [39,54], supporting the association between abundance of these adipose tissue depots and metabolic disease [55]. Similar to humans with obesity and metabolic disease, adipose tissue inflammation in cattle, mostly characterized by an increased infiltration of macrophages and high expression of pro-inflammatory adipokine genes (*LEP*, *IL6*, and *TNF*) measured by qPCR, is augmented in omental VAT compared with tailhead SAT and other VAT depots (mesenteric, intrapelvic, and perirenal) in periparturient cows [16]. Altogether, these findings suggest a beneficial metabolic role of SAT adipogenesis and a closer association of omental VAT dysfunction with metabolic disease, similarly to what is reported in humans [52,56].

Functional depot-specific differences are partly underlined by distinct endocrine functions, both in terms of the type and abundance of different adipokines. For example, leptin secretion in humans is the greatest in SAT [57,58], although circulating leptin levels are generally associated with increased whole-body adipose mass [59], suggesting that increased accumulation of VAT [60] and IMAT [61], as seen in cattle, contribute to higher leptin levels as well. Adiponectin is another adipokine with higher expression in SAT compared to other adipose depots [62]. While VAT also contributes to the secretion of adiponectin, increases in whole-body adipose mass and/or visceral adiposity are associated with decreased adiponectin levels [63]. Whether increased visceral adiposity and the associated increase in pro-inflammatory factors [64] function to inhibit adiponectin secretion or, given the anti-inflammatory functions of adiponectin [65], if the decrease in adiponectin facilitates increased expression and secretion of pro-inflammatory factors remains to be elucidated. Adiponectin expression has also been detected in IMAT, where it may act in a paracrine function to improve myocyte function [66]. Additionally, the overexpression of adiponectin in goat satellite muscle cells was observed to promote the differentiation of these cells into adipocytes, suggesting that high levels of adiponectin may support IMAT adipogenesis [67]. Depot-specific differences in the production and secretion of other adipokines, such as omentin and visfatin, which are primarily secreted from VAT [68,69], have also been observed. The pattern of adipokine secretion in periparturient dairy cattle has been previously reviewed [70]; however, there are limited studies specific to beef cattle, and our understanding of the role these adipokines play in modulating white adipose tissue and whole-body metabolic function is restricted to human and mouse models.

## 3. The Single-Cell Era: Applicability of Single-Cell/Nucleus RNA Sequencing to Decode Adipose Tissue Heterogeneity

Studies preceding the use of single-cell RNA sequencing (scRNAseq) focused on identifying cell types within adipose tissue utilizing fluorescent-activated cell sorting (FACS) based on the presence or absence of specific cell surface markers. Larger, clonally identical populations of cells were then propagated and characterized in downstream analyses. These studies were important for the initial identification of different cell types relating to adipogenesis in adipose tissue, including Lin-/Cd34+/Cd29+/Sca1+/Cd24+ ASPCs [71], Pdgfrα+ mesenchymal cells [72], Pdgfrb+ and Myh11+ mural cells [73,74,75,76,77,78,79], and mature adipocytes (ASC-1 for white; PAT2 and P2RX for brown and beige) [80]. As reviewed by Ferrero et al. [81], the use of FACS has been important in deciphering the heterogeneity of ASPCs, particularly in recent years with the development and utilization of scRNAseq applications. In addition, the work of Astori et al. [4] helped to further define the components of the stromal vascular fraction (SVF) including monocytes, granulocytes, hematopoietic stem cells, and endothelial cells. Lineage tracing via Cre-recombinase mouse models has been utilized to identify adipocyte precursors [72,76,82], and characterize mature adipocytes [83] in white adipose tissue of mice [72,76,82]. However, one limitation of the FACS- and lineage tracing-based strategies is the lack of single markers, which makes it very challenging to identify and validate ASPC heterogeneity.

With the advantage of developing next-generation sequencing (NGS)-based single-cell technologies, we can now profile adipose tissue at single-cell resolution to provide unbiased insight into its molecular composition [84]. scRNAseq was first developed by Tang et al. in 2009 to sequence the individual cell from a four-cell stage embryo [85]. Since then, significant efforts have been made on advancing various scRNAseq techniques. Among those, the droplet-based microfluidic technology is the most widely used scRNA-seq platform due to its increased throughput and more automated protocols [86,87]. In 2016, 10× Genomics first launched a commercial droplet-microfluidics-based scRNA-seq platform, the Chromium. In Chromium, cells and beads, which contain primers, cellular barcodes, and a unique molecular identifier (UMI), in separate channels are mixed in a microfluidic device and partitioned by nanoliter-size oil droplets for further single-cell cell lysis, reverse transcription, and cDNA library construction [87,88,89,90,91].

Studies have demonstrated scRNAseq as an extremely powerful tool in studying adipose tissue stromal cellular composition and plasticity. The high throughput of scRNAseq provides detailed information of the cell identity, gene expression profiles, and changes in cellular states that may arise under normal physiological and pathological conditions. Various scRNAseq studies have demonstrated the heterogeneities of the mesenchymal stem cells (MSCs), immune cells, mesothelial cells, and endothelial cells in white adipose tissues [92]. Due to their high abundancy, heterogeneity, and functional importance, many scRNAseq studies focused on ASPCs defined as cells expressing the common mesenchymal markers, *Pdgfra* and *Pdgfrb*. Moreover, owing to the power of single-cell profiling, distinct ASPC populations expressing adipogenic markers, like *Pparg*, *Lpl*, and *Cd36*, and those enriched for genes related to ECM remodeling and inflammation have been identified [92].

Most early scRNAseq studies of adipose tissue used freshly isolated intact cells, however, with some limitations. First, this strategy requires prolonged stressful enzymatic digestion, which might alter gene expression profiles of sensitive cell types. In addition, due to the prolonged sampling process of human and large animal tissues, it is very difficult to finish cell isolation immediately after sample collection on the same day. Moreover, mature adipocytes, the parenchymal cells of adipose tissue, are too large for most scRNAseq platforms. Alternatively, single-nuclei RNA sequencing (snRNAseq), which uses single nuclei, has overcome these obstacles [93,94,95]. The general workflow of sc/snRNAseq and their differences are illustrated in Figure 1. Single-nuclei RNA sequencing does not require tissue digestion (e.g., collagenase) to release individual viable cells and therefore can avoid the enzyme-induced cell-type biases, which may result in artificial transcriptional differences [96,97]. Moreover, the source of nuclei can be either fresh or frozen tissue samples, providing increased flexibility [98]. Due to the smaller size of nuclei, nuclei of all types of cells can be identified via snRNAseq, making it possible to study adipocytes and other larger cells, such as neurons. However, snRNAseq also has its limitations. One major issue is the loss of most cytosolic content, which contains most mature mRNA, leading to low numbers of genes detected and failure to detect low abundant transcripts. Another concern is the high level of ambient RNA, which must be carefully removed via additional sample and sequencing data processing [96,99]. Without the protection of cytoplasm, mRNA is more susceptible to RNase-mediated degradation, which can be an issue for samples with cells expressing high levels of RNases and must be addressed by adding sufficient RNase inhibitors [96,99].

Integrating the sc/snRNAseq data with other omics data is becoming a popular strategy to gain more comprehensive and multidimensional understanding of tissue heterogeneity. The assay for transposase-accessible chromatin using sequencing (ATACseq; Figure 1) is a powerful method for determining chromatin accessibility across the genome, an important component of the epigenome [100], and identifying potential transcription factors regulating differential gene expression [101]. In 2015, ATACseq was optimized for single-cell applications [102,103]. Single-cell and single-nuclei ATACseq facilitates the single-cell open chromatin landscaping of heterogeneous tissues like adipose tissue. It is expected that more and more studies integrating sc/snRNAseq and sc/snATACseq data to uncover the contribution of epigenetic regulation of gene expression to the heterogeneities of cells residing in adipose tissue will be published in the near future.

In the following sections, we describe recent studies utilizing sc/snRNASeq to characterize the heterogeneity of adipocyte progenitor cells, mature adipocytes, mesothelial cells, immune cells, and vascular cells in white adipose tissue. While SAT and VAT have garnered most of the research attention for their implication in metabolic diseases in both humans and livestock, recent investigations into IMAT have provided valuable insight into the mechanisms contributing to enhanced marbling in pigs and beef cattle, as well as the pathogenesis of human diseases, including muscular dystrophies and sarcopenia. Studies using sc/snRNASeq to elucidate the cellular heterogeneity of intermuscular adipose tissue are lacking but represent a novel area of study for future research endeavors in this field.

## 4. Transcriptional and Functional Diversity of Adipose Stem and Progenitor Cells (ASPC) and Mature Adipocytes

### 4.1. Adipose Stem and Progenitor Cells

Studies utilizing sc- and snRNAseq analyses have significantly advanced the field of ASPC transcriptional heterogeneity and revealed that distinct ASPCs also differentially regulate adipogenesis, inflammation, and fibrosis in a depot-specific manner, with some of these populations associated with obesity and metabolic dysregulation [19,104]. In this section, we review the main ASPC populations that have been studied in mice, humans, and livestock animals and provide a summary in Table 1.

#### 4.1.1. Mouse Models

Hepler et al. [19] identified transcriptionally and functionally distinct subpopulations of ASPCs in visceral white adipose tissue (WAT) of adult mice. In an experimental approach combining scRNAseq and FACS, the authors identified and isolated distinct subpopulations of *Pdgfrb*^+^ (CD31^−^/CD45^−^) ASPCs characterized by the differential expression of *Ly6c* and *Cd9*. *Ly6c^−^*/*Cd9*^−^/*Pdgfrb^+^* cells represented highly adipogenic visceral ASPCs, *Ly6c^+^*/*Pdgfrb^+^* cells represented fibro-inflammatory progenitors (FIPs), whereas *Ly6c^−/^Cd9^+^* represented a smaller subpopulation of mesothelial-like cells (MLCs) that lacked adipogenic capacity. While the same three populations were observed within the mesenteric and retroperitoneal depots of adult male mice, in the inguinal and anterior subcutaneous WAT depots, all *Pdgfrb*^+^ cells expressed *Ly6c*; thus, heterogeneity amongst *Pdgfrb*^+^ cells could not be discriminated based on *Ly6c* expression in these subcutaneous depots. Interestingly, after isolating and culturing these subpopulations in vitro, FIPs did not differentiate into adipocytes, displayed a pro-fibrogenic/pro-inflammatory phenotype, and exerted an anti-adipogenic effect on adipogenic ASPCs, thus highlighting cell–cell interactions that impact white adipose tissue function.

Schwalie et al. [106] had also previously identified an ASPC population with an antiadipogenic capacity. Using scRNAseq on *Lin^−^* (*CD31^−^*/*CD45^−^*/*TER119^−^*) *CD29^+^*/*CD34^+^*/*SCA1^+^* cells from the SAT SVF of transgenic adult mice, this group identified three populations of ASPCs: The two major ones made up over 90% of all ASPCs and expressed stem cell-specific markers (*Cd34* and *Ly6a*) or a pre-adipogenic (*Fabp4*, *Pparg* and *Cd36*) gene profile. The third ASPC population was characterized by *F3* (encoding CD142) and *Abcg1* and had a decreased propensity to form adipocytes compared with the other ASPC populations. Notably, isolated subcutaneous C*D142^+^*/*ABCG1^+^* ASPCs negatively regulated the adipogenic capacity of other ASPCs, being named as ‘adipogenesis-regulatory’ cells or A-regs [106]. Merrick et al. [21] also identified a CD45^−^ ASPC population in mice SAT that expressed *F3* (CD142), *Clec11a*, and *Fmo2* [106]. However, differently to what was demonstrated by Schwalie et al. [106], isolated CD142^+^ ASPCs were able to differentiate into lipid laden adipocytes in vitro in both conditions, complete adipogenic medium or minimal insulin medium, and did not demonstrate anti-adipogenic properties. These differences between studies may reflect discrepancies in the experimental conditions, age, and stage of development of the animals. For instance, mice utilized in Schwalie’s work were 8–11 weeks old, while Merrick’s work used 12-day-old mice as an attempt “to capture the continuum of cell states spanning differentiation”. In addition to a CD142^+^*Clec11a^+^* ASPC population, Merrick’s work also identified and functionally characterized additional ASPC populations in mice inguinal SAT CD45^−^ SVF (the presence of which was further validated in mice axillar SAT, interscapular brown adipose tissue, and epididymal VAT of adult mice) [21]. *Dpp4^+^* ASPCs were classified as “Interstitial Progenitor Cells” due to their origin in the adipose tissue reticular insterstitium and seemed to provide a source of both *Icam1^+^* and *Cd142+* ASPCs. *Icam1^+^* ASPCs were classified as committed preadipocytes, as they also expressed several adipocyte identity genes such as *Dlk1* (Pref1), *Pparg*, *Fabp4*, and *Cd36* and showed abundant lipid droplet accumulation in vitro. Both *Dpp4^+^* and *Icam1^+^* ASPCs also expressed canonical mesenchymal progenitor markers, such as *Cd34*, *Pdgfra*, *Ly6a* (Sca1), and *Thy1* (CD90), and differentiated into lipid laden adipocytes in vitro. Interestingly, transforming growth factor-beta maintained *Dpp4^+^* cell identity and inhibited adipogenic commitment of *Dpp4^+^* and *Cd142+* cells [21].

A follow-up study by Stefkovich et al. [118] confirmed that *Dpp4^+^* progenitors contribute to adipogenesis not only in the murine subcutaneous WAT, but also in omental and retroperitoneal visceral depots. Our lineage tracing also identified *Dpp4^+^* and *Icam1^+^* cells in the *Tcf21* lineage VAT ASPCs [101]. *Tcf21* was found to be a VAT-specific gene and not expressed on subcutaneous WAT or brown adipose tissue. It inhibits adipogenic differentiation of VAT ASPCs at least partially through promoting the expression of *Dlk1*, an anti-adipogenic gene, in the *Icam1^+^* population. A general reduction in adipogenic activity in *Tcf21* lineage ASPCs as the mice become increasingly mature was observed. Mechanistic studies using bulk- and scRNAseq identified increased hypoxia-related gene expression in *Dpp4^+^ Tcf21* lineage ASPCs located in mesothelium and activation of inflammatory and fibrotic programming in the interstitial *Icam1^+^* population [101]. In contrast to mouse models, our recent snRNAseq analysis of abdominal SAT and omental VAT from dairy cows revealed a negligible gene expression of *DPP4* and *F3* (CD142) [33], which may suggest a species-specific origin and development of adipocytes.

Further work elucidating the transcriptional heterogeneity of ASPCs has recognized a unique subpopulation of *PDGFRA^+^* ASPCs called fibro/adipogenic progenitors (FAPs) with both fibrotic and adipogenic capacity [119]. Recently, Garritson et al. [114] identified seven different FAP subpopulations in both human and mouse VAT including uncommitted adipose progenitors (I*i16^+^*, *Sema3c^+^*, *Osr2^+^*), committed adipose progenitors (*Icam1^+^*, *Cebpb^+^*), pro-fibrotic progenitors (*Vcan^+^*, *Mfap^+^*, *Htra3^+^*) and a stem cell population expressing adipogenic inhibitors (*Thy1*/Cd90*^+^*, *Cthrc1^+^*). A general population of FAPs was also identified in VAT of mice by Sárvári et al. [95], characterized by high expression of *Col1a1*, *Pdgfra*, and *Dcn*. In a study evaluating both depot- and sex-specific differences in ASPC heterogeneity in mice, Shan et al. [110] identified two distinct *Pdgfrb^+^* ASPC subpopulations in SAT (*Dpp4^+^* and *Dpp4^−^*) and VAT (*Ly6c^−^*, *Cd9^−^*) as well as a subpopulation of VAT-specific fibro-inflammatory precursors (FIPs) (*Ly6^+^*). Functional analysis confirmed the increased adipogenic capacity of the ASPC subpopulations compared to FIPs, as well as highlighted sex-specific differences in ASPC adipogenesis, with SAT ASPCs in female mice exhibiting greater PPARγ activation than the same subpopulation of ASPCs from male mice. Further investigation also revealed that both the AhR and glutathione pathways are implicated in the functional differences between ASPCs and FIPs, with reduced AhR signaling in FIPs contributing to a pro-inflammatory transcriptional response, and increased *Gstm1* expression in ASPCs supporting increased adipogenic potential.

Burl et al. [105] applied scRNAseq to identify the effects of beta-3-adrenergic receptor (ADRB3) activation on epididymal and inguinal WAT ASPCs heterogeneity. Analysis of Lin- cells resulted on the identification of four major ASPC expressing *Pdgfra* and *Ly6a*, which demonstrated to be in different stages or development and adipogenic differentiation. ADRB3 activation shifted the expression profiles of the major ASPC subtypes (ASC 1 and ASC 2) in both adipose depots and induced the appearance of two new clusters of differentiating ASPC (*Cebpa+*, *Agpat2+*, *Dgat2+*, *Plin1+*, *and Adipoq+*) and proliferating ASPC. However, the differentiation of these cells into beige/brown adipocytes could not be evaluated through single-cell analysis. This study was one of the first studies to use scRNA sequencing to elucidate ASPC heterogeneity, adipogenic niches, and transcriptional responses of individual ASPCs to ADRB3 activation.

In general, studies of ASPC heterogeneity in murine models have highlighted the transcriptional and functional heterogeneity of visceral and subcutaneous WAT ASPCs and demonstrated the ability of distinct stromal cell populations to crosstalk and modulate adipogenic, fibrotic, and inflammatory functions in WAT.

#### 4.1.2. Human Models

Raajendiran et al. [108] identified three major ASPC populations utilizing scRNAseq of human abdominal SAT SVF based on the differential gene expression of CD34 (negative, low, and high) in CD31^−^/CD45^−^/CD29^+^ cells. After FACS isolation of the distinct ASPCs from human VAT and abdominal and gluteal SAT, cells were induced to differentiate into adipocytes for metabolic characterization in vitro. All three ASPC populations demonstrated similar adipogenic capacity; however, CD34^high^ adipocytes exhibited enhanced capacity to accumulate and release lipids compared with CD34^−^ or CD34^low^, while CD34^−^ acquired a beige-like adipocyte profile, which was more abundant in gluteal SAT. Notably, human subjects with type 2 diabetes had a decreased proportion of CD34^−^ ASPCs and an increased proportion of CD34^high^ in both VAT and abdominal SAT [108], which may partly underly the defects on adipose tissue lipid metabolism associated with obesity and type 2 diabetes.

In the work of Emont et al. [93], six distinct *PDGFRA^+^* ASPC subpopulations were identified in human SAT and VAT samples including *ALDH1A3^+^* ASPCs that resemble the multipotent progenitor cells identified by Merrick et al. [21], and *EPHA3^+^* ASPCs similar to the anti-adipogenic A-regs identified by Schwalie et al. [106]. Among the ASPCs identified by Emont et al. [93], three subpopulations (*CEBPD^+^*, *EPHA3^+^*, and *SGCZ^+^*) were more abundant in SAT compared to VAT, with *EPHA3^+^* and *SGCZ^+^* ASPCs further increased in subjects with higher BMI. In contrast, two ASPC subpopulations (*FGF10^+^* and *PDE4D^+^*) were more abundant in VAT compared to SAT.

Seven CD45^−^/CD34^+^/CD31^−^ ASPC subpopulations were identified by Vijay et al. [20] in human SAT and VAT, with three subpopulations found primarily in SAT, three subpopulations found primarily in VAT and one subpopulation evenly distributed between the two depots. Further analysis of the ASPCs found primarily in SAT identified four distinct transcriptional profiles, including pre-adipocyte/adipose stem cells (*MGP^+^*/*APOD^+^*/*CXCL14^+^*/*WISP2^+^*), mature adipocyte progenitor cells (*APOE^+^*/*FABP4^+^*/*CEBPB^+^*/*CD36^+^*), fibrosis and ECM-associated cells (*COL3A1^+^*/*COL6A3^+^*/*COL1A1^+^*/*COL6A1^+^*), and pro-inflammatory cells (*CCL5^+^*/*CD3E^+^*/*IL7R^+^*/*IL32^+^*). Among all the ASPC subpopulations in SAT, it was found that *GPX3* expression was significantly higher in samples from healthy subjects compared to those with type 2 diabetes. The opposite was observed for *WISP2*, which was higher in samples from subjects with type 2 diabetes. Further analysis of the VAT ASPCs identified six unique transcriptional profiles, three of which were similar to the pre-adipocyte populations described in SAT, while the other three subpopulations were marked by an increased expression of *MSLN* and were classified as mesothelial cells. These findings highlight the depot- and metabolic disease-specific profiles of ASPCs in humans.

Both ASPCs and preadipocytes were identified in the analysis of human SAT performed by Hildreth et al. [109], with ASPCs characterized by high expression of *PRG4*, *DKK1*, and *PI16*. Preadipocytes had high expression of *PDGFRA* but were distinguished based on their high expression of *CXCL14* and *GPC3* and lower expression of the other ASPC marker genes. Notably, there was a positive correlation between patient BMI and the abundance of ASPCs; however, a negative correlation was observed between patient BMI and the abundance of preadipocytes, highlighting the relationship between obesity and changes in adipose tissue function.

Strieder-Barboza et al. [112] used snRNAseq to define adipose tissue heterogeneity in abdominal SAT and omental VAT in human subjects going through bariatric or elective surgery. Three distinct subpopulations of ASPCs were identified based on expression of *PDGFRA* and *PDGFRB*. The ASPC1 population, named “inflammatory mesothelial-like ASPC” (IM-ASPC), was present in VAT, but not in SAT, and exhibited a unique expression of *TM4SF1*, high expression of mesothelial cell signature genes (*WT1*, *MSLN*, *CLDN1*, and *KRT19*), and an enrichment for multiple inflammatory gene pathways including *IL-1*, *TLR*, *TNF*, and hypoxia signaling. Notably, the IM-ASPC gene profile was similar to the *Dpp4+* ASPC described by Merrick et al. [21], the mesothelial-like cells (MLC) reported by Hepler et al. [19], and the *MSLN+* VP1/VP3 revealed by Vijay et al. [20]. IM-ASPCs were also similar to the PDGFRa/CD9^hi^ ASPCs described by Marcelin et al. [120], which contribute to adipose tissue fibrosis in obesity. The ASPC2 population was the most prominent ASPC in SAT and had a fibro-adipogenic profile (FAP-ASPC) with an increased expression of collagens and proteoglycans, such as *DCN* and *LUM* [112]. FAPs have the capability to differentiate into adipocytes or activated fibroblasts increasing extracellular matrix deposition [121,122]. Accordingly, FAP-ASPC was enriched in gene pathways relating to ECM remodeling and organization and in collagen production [112]. The FAP-ASPC gene signature was similar to *Cd142+* ASPC, as described by Merrick et al. [21] in mice, and the human CFD+ SP1/SP4/VP4 populations reported by Vijay et al. [20] that were associated with type 2 diabetes. The smallest ASPC3 population had high expression of smooth muscle markers and had a similar gene profile that overlaps with *Icam1+* ASPC [21]. After FACS isolation and adipogenic differentiation in complete and minimal insulin medium in vitro, both IM-ASPC and FA-ASPC formed lipid laden adipocytes. However, FAP-ASPC had an increased lipid accumulation, glycerol release in response to forskolin, and augmented expression of adipogenic markers including *PPARG*, *PLIN1*, and *ADIPOQ* compared with IM-ASPC [112]. These results demonstrate that ASPCs transcriptional differences are maintained and translated into distinct functional profiles after in vitro differentiation, and seem to agree with previous studies in human and mice models that have tested the metabolic function of distinctive ASPC subpopulations in vitro [19,21,93,106].

Single-cell RNA sequencing studies reporting IMAT ASPC heterogeneity are less frequent. However, in human skeletal muscle, Fitzgerald et al. [115] recently identified three *PDGFRA^+^*/*CD34^+^*/*DCN^+^* FAP subpopulations, in which one specific *MME^+^* FAP subpopulation with high adipogenic capacity appeared to be unique to the skeletal muscle tissue, while *GPC3^+^* FAPs shared marker genes with both the muscle-specific *MME^+^* cells and the *CD55^+^* FAPs that have been detected in other adipose tissue depots.

#### 4.1.3. Livestock Models

The use of scRNAseq and other multi-omics technologies is still very limited in livestock species, especially on the study of adipose tissue function and biology (Figure 2). However, a few studies have performed sc/snRNAseq in adipose tissue and muscle samples in beef and dairy cattle, chicken, and pigs, thus revealing an interesting heterogeneity of ASPCs that seems to vary with anatomical location of adipose tissue as well as the species and breed of animals [117].

Recently, we have used snRNAseq analysis to study depot-specific cellular heterogeneity of abdominal SAT and omental VAT in dairy cows [33], which identified three ASPC subtypes across SAT and VAT. The analysis of DEGs identified ASPC1 as an adipogenic subtype with a high expression of *PPARG*, as seen in committed adipocyte precursors [104]. Adipogenic-ASPCs were enriched for lipid metabolism, homeostasis, and biosynthesis genes and corresponded to 50% of all ASPC across VAT and SAT. Notably, adipogenic-ASPCs were decreased 2-fold in VAT compared to SAT, suggesting depot-specific shifts in frequency of ASPCs subtypes. ASPC2 and ASCP3 were identified as fibro-adipogenic progenitors (FAP) expressing high *PDGFRA*, *LAMA2*, *FBN1*, and *VCAN* [120]. Accordingly, pathways of collagen metabolism and fibril organization, and ECM structure and organization were activated in FAPs. Similarly to previous data in mice and human skeletal muscle [123,124], we identified multiple FAP subtypes: ASPC2 showed a classic-FAP profile, expressing high *ADAM* genes, *BMP1*, *EBF1*, *FGFR1*, and *TGFB* receptors, which have been recently linked to low marbling in beef cattle [125], while ASPC3 had a fibrogenic-FAP profile with high expression of fibrosis-associated genes [126] (collagens, *FN1*, *DCN*, *FBLN1*, *MMP2*, *LUM*, *SPARC*, and *LOX*). This profile overlaps with human VAT ASPCs, which are positively correlated with insulin resistance [20]. Fibrogenic-FAPs also showed a pro-inflammatory potential with upregulation of *CCL2*, *CXCL3*, *C3*, *C1S*, and *PTGS2*.

Our group has also recently identified the heterogeneity of ASPCs in beef cattle muscle tissue. We report that bovine muscle ASPCs or FAPs are composed of multiple distinct but related populations, including a perimysial fibrogenic, an endomysial adipogenic, and a transitional population, all derived from a common less-committed population [116]. Besides the expression of well-known proadipogenic genes, such as *PPARG* and *BMP* family genes, adipogenic FAPs also expressed elevated levels of *CFD* and non-fibrillar collagen genes, such as *COL4A1*, in contrast to the enriched expression of profibrotic genes, including *POSTN*, *TGFBR3*, and fibrillar collagen genes. Interestingly, more prominent adipogenic programming and fibrogenic programming were identified in FAPs of Wagyu and Brahman cattle, respectively, and likely contribute to the drastic differences in beef quality between the two breeds. The specific early expression of *CFD* in FAPs of young Wagyu calves likely predispose them to adipogenesis. Moreover, similar adipogenic FAPs were identified in humans and non-human primates but not rodents, suggesting common transcriptional programming of FAPs among larger mammals [116]. *PDGFRA^+^* FAPs have also been identified in IMAT of swine [117] and beef cattle cultured satellite cells [127] and remain an important focus of research given their potential for modulating intramuscular adipose development.

### 4.2. Mature Adipocytes

Until recently, only two major types of adipocytes were recognized in humans based on adipose tissue type: white (UCP1-) and brown (UCP1+) adipocytes. Similarly to the discovery of a repertoire of transcriptionally and functionally distinct ASPCs in WAT, single-cell technologies have facilitated the characterization of distinct adipocyte subtypes that are not readily distinguished by their morphology, but play distinct metabolic functions varying across depots and health status in human and animal models [33,93,104].

A limitation of scRNAseq analysis of adipose tissue is that mature adipocytes are missing in datasets as they cannot be sorted by microfluidics because of their large size and high buoyancy (Figure 1). To overcome this, several research groups have developed methods to isolate and sequence single nuclei extracted from adipose tissues [33,93,112]. In this section, we describe distinct mature adipocyte subtypes that have been reported in human, mice, and livestock models.

#### 4.2.1. Mouse Models

A few studies using scRNAseq of adipose tissue have identified clusters of small, differentiating adipocytes that were mixed within the SVF prior to single-cell suspension preparations [20,21,105]. For example, Merrick et al. found a small group of cells (Group 7), which were classified as “adipocytes” because they lacked expression of progenitor markers and expressed high levels of *Adipoq*, *Plin1*, and *Car3*, which are known signature genes for mature adipocytes [21]. Similarly, Burl et al. [105] reported a group of newly formed adipocytes or differentiating ASPCs, which did not contain enough intracellular lipids to be buoyant and were copurified with the adipose tissue SVF.

Using snRNAseq analysis, Emont et al. [93] identified six transcriptionally distinct mouse mature adipocyte (mAd1-6) subtypes in inguinal and perigonadal WAT, all expressing *Adipoq*, which were clearly distinct from human adipocyte subtypes and did not demonstrate a depot-enrichment. Notably, changes in the abundance of specific adipocyte subtypes were dependent on a high-fat diet (HFD). An HFD reduced the proportion of adipocyte subtypes expressing high *Ces1f* (mAd1) and *Apoe* (mAd3) and increased the abundance of high *Cacna1a* (mAd4) adipocytes and the lipogenic adipocyte subtype characterized by high *Prune2* (mAd5). Functional analysis revealed an association of HFD-associated adipocytes with pathways related to hypoxia, cytoskeleton remodeling and inflammation, and insulin resistance. Notably, no thermogenic mouse adipocyte subtype was identified. However, when excluding HFD-associated adipocyte populations from the analysis and sub-clustering mAd1, the authors observed the presence of three adipocyte subtypes expressing thermogenic beige adipocyte markers, such as *Prdm16*, *Ppargc1a*, *Ucp1*, and *Cidea*, which were enriched in inguinal vs. perigonadal fat. In summary, the Emont et al. [93] results demonstrate diet-dependent effects on adipocyte abundance and profiles in distinct adipose tissue depots. A group adipocytes was also identified in VAT of mice by Sárvári et al. [95], characterized by high expression of *Lipe*, *Plin4*, and *Pparg*. Interestingly, the abundance of adipocytes was decreased in VAT samples from obese mice compared to lean mice, partially explained by a simultaneous increase in the abundance of immune cells in the same tissue samples [95].

#### 4.2.2. Human Models

In a recent study by Emont et al. [93], seven distinct subpopulations of *ADIPOQ^+^* mature adipocytes were identified, with strong depot-specificity. Of the seven adipocyte subpopulations, four were found primarily in SAT (*GALNT13^+^*, *PNPLA3^+^*, *GRIA4^+^*, and *AGMO^+^*), while two were found primarily in VAT (*TNFSF10^+^* and *EBF2^+^*). Further analysis of each of the adipocyte subpopulations highlighted the potential link between adipocyte diversity and depot-specific functional differences, with *PNPLA3^+^* and *GRIA4^+^* adipocytes from SAT exhibiting more adipogenic profiles and *EBF2^+^* adipocytes from VAT exhibiting a thermogenic profile. An additional adipocyte subpopulation, *PGAP1^+^*, was evenly distributed between SAT and VAT and exhibited a more lipolytic expression profile. Interestingly, proportions of *GRIA4^+^* and *AGMO^+^* adipocytes were negatively correlated with BMI, whereas the proportion of *PGAP1^+^* adipocytes was positively correlated with BMI suggesting a link between obesity and changes in specific adipocyte subtype function.

Similarly to the Rosen group’s methodology [93], Lumeng’s group performed snRNAseq analysis in human abdominal SAT and omental VAT and reported seven transcriptionally distinct mature adipocyte populations [112]. Interestingly, the proportion of mature adipocytes in SAT was twice as high as in VAT and enriched for ECM, integrin, and adipogenic gene pathways genes compared to an enrichment for mesothelial and nuclear receptor genes in VAT adipocytes. The most frequent adipocyte subtype (C0) expressed *ITGB1/CD29*. Additionally, the group identified adipocyte subtypes with high expression of multiple markers of FA-ASPCs (*DCN*, *F3*, *LUM*, *NEGR1*) and IM-ASPCs (*TM4SF1*, *DPP4*), thus suggesting that distinct adipocyte types retain expression of signature ASPC marker genes. Unfortunately, the limited sample size did not allow the authors to assess the associations between specific adipocyte types and obesity and type 2 diabetes. However, the authors reported that VAT adipocytes from lean subjects were enriched for fatty acid catabolic genes and genes relating to corticosteroid responses, compared with VAT adipocytes from subjects with obesity, thus suggesting alterations on adipocyte function depending on BMI.

In their study of human SAT, Whytock et al. [111] described adipocytes (*DGAT2*, *PLIN1*, *LEP*, *ADIPOQ*, *PPARG*, *LIPE*, *FABP4*, *SAA1*) as the most abundant cell type, with distinct transcriptional differences between three subpopulations representing different stages of adipocyte maturation. One of the subpopulations were considered as early adipocytes, characterized by high expression of early adipogenic genes (*CEBPA*, *STAT5A*) and genes associated with lipid droplet formation (*SYNGR2*, *PLIN4*, *CES1*). A second subpopulation of adipocytes were considered as mature glycolytic adipocytes, characterized by a high expression of glucose homeostasis (*MLXIPL*, *PIK3CA*, *HK2*) and insulin signaling genes (*IRS1*, *IRS2*, *PIK3CA*). The third subpopulation of adipocytes were characterized by the highest expression of mature adipocyte markers (*LEP, ADIPOQ, PPARG, SAA1*) and were considered fully developed adipocytes [111].

A recent manuscript integrated single cell transcriptome data from 10 distinct studies, from which seven snRNAseq data sets were analyzed for adipocyte heterogeneity in subcutaneous, omental, and perivascular adipose tissue depots [128]. The study revealed that adipocytes constitute ±20% of human white adipose tissue cell population but present a lower degree of heterogeneity compared to other cell types and a limited number of reproducible adipocyte marker genes among studies. Surprisingly, adipocyte signature genes (e.g., *LEP*, *PLIN4*, *SAA1*, *RBP4*) were absent or lowly expressed in snRNAseq datasets. Yet, genes involved in lipid metabolism, such as *ABCD2*, *ACACB*, *CD36*, *DGAT2*, *GPAM*, *HACD2*, and *LPL* seem to be found in most of identified adipocyte subtypes.

#### 4.2.3. Livestock Models

In livestock species, only limited data are available regarding mature adipocyte heterogeneity. Consistent with ASPC diversity, our study utilizing snRNAseq analysis of dairy cows SAT and VAT identified four distinct mature adipocyte subtypes across depots that selectively expressed *ADIPOQ* and/or *LEP* [33]. Our results demonstrated that ADIPOQ and LEP protein were not expressed by all adipocytes, implying functional differences or distinct lineages of adipocytes. We observed that one population of adipocytes (AD1) had an adipogenic-ASPC1-like profile with a high expression of classical lipid synthesis regulators (*PPARG*, *FASN*, *GPAM*, *INSG1*, and *ZNF106*) and downregulation of ECM genes. In contrast, AD2 and AD3 had an FAP-like profile expressing *PDGFRA*, ECM genes (*DCN*, *FBN1*, *LAMA2*, and *COL1A1/A2*), and genes associated with lipid synthesis, such as *ADIRF*, *SCD*, *FABP4*, *AGPAT2*, *APOE*, and *LPL*. AD4 had a unique profile with high *LPIN1*, *CSF1*, and collagens expression. These results suggest that distinct ASPC subpopulations give rise to distinct adipocyte types in bovine adipose tissue; however, the functional characterization of these subpopulations needs to be established.

Single-cell analyses have also been performed in other livestock species. In breast muscle of chickens at 5 and 100 days old, two different mature adipocyte populations were identified at day 5, and one mature adipocyte population was identified at day 100 [129]. Canonical adipocyte marker genes such as *ADIPOQ* and *FABP5* were characteristic of one of the day-5 adipocyte populations, while the other population exhibited increased expression of other genes associated with lipid accumulation including *GPX3*, *APOA1*, *COL1A1*, and *COL6A3*. The adipocyte population identified at day 100 exhibited a similar expression profile to the adipocyte subpopulations identified at day 5, although made up a smaller percentage of total identified cells. Evaluating adipocyte populations in longissimus dorsi samples of swine with high and low degrees of intramuscular fat, Wang et al. [117] identified three (*DGAT2^+^/SCD^+^*, *FABP5^+^/SIAH1^+^*, *PDE4D^+^/PDE7B^+^*) different mature adipocyte subpopulations. Among these subpopulations, the *DGAT2^+^/SCD^+^* and *FABP5^+^/SIAH1^+^* adipocytes made up a higher percentage of total adipocytes in the samples from animals with higher intramuscular fat. These findings highlight adipocyte diversity in IMAT in livestock and suggest that changes in the transcriptional profile of adipocytes contribute to the development of this adipose tissue depot. Future studies similar to the work of Li et al. [129] that evaluate adipocyte transcriptomic profiles in IMAT at different livestock production stages will be necessary for understanding how IMAT develops and how this process may be modulated to improve animal meat products.

## 5. Transcriptional Diversity of Adipose Tissue Immune Cells

The network of immune cells in white adipose tissue represents an important link between metabolic and immune function and these cells are implicated both in the prevention and pathogenesis of metabolic disease and inflammation. Recent sc/snRNASeq approaches have provided critical insight into the specific types of immune cells present in adipose tissue as well as transcriptional changes that occur in these cells during different metabolic conditions.

### 5.1. Mouse Models

In their study on β3-adrenergic stimulation in mice, Burl et al. [105] identified populations of macrophages (*C1qa*, *Trem2*, *Adgre1*, *Spp1*) and natural killer and T cells (NKT; *Nkg7*), as well as a population of mixed macrophages and dendritic cells (*Cd74*) in epididymal VAT. Notably, β3 stimulation resulted in an increase in the abundance of macrophages and a decrease in the abundance of NKT cells in adipose tissue. Additionally, the authors described a subpopulation of lipid-associated macrophages characterized by a high expression of *Spp1*, *Cd36*, *Fabp5*, *Lpl*, and *Lipa* and suggested to be involved in the clearance of dead adipocytes and stimulation of adipogenesis by new adipocytes.

Nine distinct immune cell subpopulations were identified in VAT (epididymal adipose tissue) from obese and lean mice by Sárvári et al. [95], six of which were classified as macrophages (*Adgre1*, *Lyz2*, *Ccl6*), with one subpopulation of dendritic cells (*Cd244a*, *Cd209a*), one subpopulation of T cells (*Skap1*), and one subpopulation of B cells (*Ms4a1*, *Cd79a*, *Cd79b*). Among macrophages, the authors identified subpopulations of perivascular macrophages (*Mrc1*, *Lyve1*, *Cd163*), lipid-associated macrophages (*Lpl*, *Trem2*, *Cd9*), non-perivascular macrophages (*Fcrls*, *Ear2*), collagen-expressing macrophages (*Col5a2*, *Tgfbr3*, *Col3a1*), proliferating lipid-associated macrophages (*Pola1*, *Kif11*, *Kif15*) and regulatory macrophages (*Prg4*, *Tgfb2*, *Ltbp1*). Notably, the authors found a dramatic increase in lipid-associated and proliferating lipid-associated macrophages in obese mice, and a shift in the transcriptional profile of perivascular and non-perivascular macrophages towards increased expression of lipid metabolism genes (*Pparg*, *Lpl*) compared to lean mice.

### 5.2. Human Models

Strieder-Barboza identified macrophage (MAC; *MERTK*, *CD163*), dendritic cell (DC; *CD1D*, *FLT3*), T cell, and NK cell (*SKAP1*) populations that were similarly distributed across human abdominal SAT and omental VAT [112]. Four major MAC subtypes were identified including lipid-associated macrophages (LAM) expressing high *ITGAX*, *TREM2*, *CD9*, and *CD52*, and two *MRC1*/CD206+ MAC subtypes expressing marker genes of resident macrophages such as *F13A1*, *PDGFC*, and *LYVE1*, as previously reported [130]. Another resident macrophage was identified by the lower *MRC1* expression and the unique *TIMD4* expression [131]. Emont et al. [93] also identified a range of immune cells in human SAT and VAT samples including macrophages (*MAFB*, *CYBB*), monocytes (*CYBB*, *CSF3R*), dendritic cells (*FLT3*), mast cells (*CPA3*), neutrophils (*CSF3R*), B cells (*MS4A1*), NK cells (*KLRD1*), and T cells (*IL7R*). While Emont et al. [93] included mouse adipose tissue samples in their analysis, they did not identify major differences between mouse and human immune cells.

Analyzing SAT and VAT samples from human patients undergoing bariatric surgery, Vijay et al. [20] found a similar panel of immune cells including naïve T cells (*IL7R*), activated T cells (*CCL5*, *IL32*), NK cells (*NKG7*), as well as a subpopulation of dysfunctional T cells characterized by high expression of the metallothioneins *MT1F* and *MT1G*. Macrophages (*CD68*) were also identified in this study, including a subpopulation of lipid-associated macrophages characterized by a high expression of lipid metabolism-related genes (*LIPA*, *LPL*, *CD36*, *FABP4*), as well as subpopulations of pro-inflammatory (*CXCL3*, *CXCL8*, *IL1B*) and anti-inflammatory (*FOLR2*, *KLF4*) macrophages. Dendritic cells, characterized by a high expression of *HLA* genes, and B cells (*IGKC*, *CD79A*, *CD37*) were also identified in these adipose tissue samples [20]. Resident (*CD14*, *CD8*, *CD163*) and pro-inflammatory (*ITGAX*, *CD86*, *TREM2*) macrophages were also described by Whytock et al. [111] in human SAT. Immune cells were identified in the integrated dataset of human white adipose tissue samples described by Massier et al. [128], including B cells (*MS4A1*), mast cells (*CPA3*), T, NK, and NKT cells (*CD3D*), and a large group of cells including macrophages, dendritic cells, and monocytes (*MRC1*, *CD11c*, *MRC1*, *FABP4*). Innate lymphoid cells (ILCs) have only recently been characterized as important components of the innate immune system and are typically associated with their roles at mucosal barriers; however, Hildreth et al. [109] identified a subpopulation of ILCs in human SAT, characterized by high expression of *KLRB1*, *KIT*, *CD200R1*, and *CCR6*. Supporting the findings of Sárvári et al. [95] and Vijay et al. [20], Hildreth et al. [109] also identified a subpopulation of lipid-associated macrophages (*TREM2*, *CD9*, *LPL*), that were in greater abundance in SAT samples from obese patients compared to lean patients.

Analyzing fatty infiltration in human skeletal muscle, Fitzgerald et al. [115] identified a general population of macrophages (*MRC1*, *C1QA*) as well as a subpopulation of inflammatory macrophages (*CD68*, *PECAM1*), but did not describe the presence of additional immune cell populations.

### 5.3. Livestock Models

Our recent snRNAseq data on dairy cows’ abdominal SAT and omental VAT revealed populations of macrophages, and natural killer and T cells, and a significant increase in the abundance of these immune cell populations in VAT vs. SAT, specifically on the proportion of lipid-associated macrophages (LAM) [33]. Macrophages (MAC) expressing *MRC1* and *MSR1* corresponded to nearly 10% of all adipose tissue nuclei and were the most heterogenous cell type in the adipose tissue of dairy cows. We identified five distinct MAC subtypes, including two subtypes of LAMs (*FABP4*, *LPL*, *CD36*, *FASN*, *CD9*), perivascular MACs, M2-like MACs (*CD163*, *CD206*), and a MAC subtype enriched for complement genes (*C3*, *CFI*, *CFB*) and *S100A12* and *S100A8*, suggesting these cells were likely monocytes or differentiating macrophages [20]. Dairy cow adipose tissue MAC subpopulations were similar to the ones identified via snRNAseq across human SAT and VAT [112]. Notably, we also identified a SAT-specific MAC subtype characterized by *ABL1*, *SPTBN1*, *ZBTB16*, and *ASAMTSK3.* In addition to MAC, our snRNAseq data also revealed a cluster of cells that were enriched for gene markers of T cells (*CCL5*, *CD3E*, *CD2*, *CD247*, *CD52*) and natural killer cells (*NKG7*, *CTSW*) comprising nearly 3% and 1.3% of total nuclei in VAT and SAT, respectively. Our data on the immune cell abundance and heterogeneity in the SAT and VAT of dairy cows highlighted depot specificities and potential distinct functional roles for immune cell subtypes, particularly macrophages. However, the specific roles of MAC and NKT populations in the SAT and VAT of dairy cows and how they underly health and disease statuses are yet to be elucidated.

In other animal models, Li et al. [129] did not detect any immune cells in the breast muscle of chicken at either 5 and 100 days old, whereas a distinct population of myeloid-derived cells (*MRC1*) and a population of general immune cells (*CD3E*) were identified in the IMAT of pigs [117].

## 6. Transcriptional Diversity of Adipose Tissue Vascular Cells

The vascular system plays an important role in adipose tissue, providing nutrients and oxygen to support adipose tissue development and expansion. Single-cell and single-nuclei RNA sequencing have significantly improved our understanding of the complexity of endothelial cells in adipose tissue, highlighting depot-specific differences in vascular and lymphatic networks. According to a recent study, adipose tissue vascular cells are broadly divided into four major groups, including vascular endothelial cells, lymphatic endothelial cells, vascular smooth muscle cells, and pericytes [128]. In this section, we will discuss current data on all four groups of vascular cells divided by model type.

### 6.1. Mouse Models

Endothelial cells were identified in VAT of β3-adrenergic stimulated mice by Burl et al. [105], characterized by a high expression of *Cldn5*, *Aqp7*, *Cdh5*, *Kdr*, *Aqp1*, and *Flt1*. In addition, the authors identified a population of cells expressing smooth muscle and pericyte markers such as *Myl9* and *Steap4*. Three clusters of endothelial cells were identified by Sárvári et al. [95] including vascular cells (*Vegfc*, *Vcam1*, *Vwf*), lymphatic cells (*Lepr*, *Ccl21a*, *Lyve1*) and endothelial progenitor cells (*Cdh5*, *Kdr*, *Flt1*) in VAT samples from lean and obese mice, with an increase in the expression of pro-inflammatory cytokines (*Cxcl13*, *Ccl8*, *Ccl9*, *Ccl6*) among these cells in the obese animals. Mice were also included in the study performed by Emont et al. [93]; however, the main subpopulations identified were conserved between humans and mice.

### 6.2. Human Models

The recent study by Massier et al. [128], integrating 17 distinct white adipose tissue sc/snRNAseq databases, highlights that vascular cells comprise nearly 15% of all human adipose cell types and present angio- and adipogenic gene profiles. Notably, Strieder-Barboza et al. report that more than 30% of all SAT cells are composed of vascular cells [112], including lymphatic (high *PROX1* and no *VWF*) and venous (*NR2F2*, *EPHB4*, and *SNTB*) endothelial cells, a putative endothelial cell progenitor subpopulation expressing low *VWF* and high *EFNB2*, and pericytes (*NOTCH4*, *DLL4*, *JAG2*, *HEY1*). Similarly to what was reported by Massier et al. [128], vascular cells expressed genes involved in lipid metabolism, such as *PPARG*, *FABP4*, and *ZNF423*, with these cells containing adipogenic potential [112].

Working with their large dataset, Massier et al. [128] identified multiple unique vascular cell clusters including capillary cells (*CD36*, *BTNL9*), venous cells (*EPHB6*, *ACKR1*), artery cells (*EFNB2*), pericytes (*PDGFRB*, *CSPG4*, *RGS5*), lymphatic cells (*PROX1*, *LYVE1*), and smooth muscle cells (*ACTA2*, *MYH11*, *TAGLN*) [128]. Similar types of cells were identified in human SAT and VAT by Emont et al. [93], including lymphatic endothelial cells with high expression of *PROX1*, pericytes (*STEAP4*), and smooth muscle cells (*MYOCD*). General endothelial cells (PECAM1, CDH5) were also identified in human skeletal muscle by Fitzgerald et al. [115] and by Hildreth et al. [109] in another set of human SAT samples.

Three clusters of endothelial cells, characterized by a high expression of *GNG11* and *SEPW1*, were identified in the human SAT and VAT samples described by Vijay et al. [20] including one subpopulation that expressed classical endothelial markers such as *VCAM1*, *ICAM1*, *PECAM1*, and *VWF*. Another subpopulation was identified as lymphatic endothelial cells based on the high expression of *LYVE1*. Notably, a subpopulation of endothelial cells with high expression of lipid metabolism associated genes (*FABP4*, *CD36*) was also identified and suggested to be a population of microvascular endothelial cells. A similar subpopulation of lipid-related endothelial cells (*FABP4*, *CD36*) was identified in the study of Whytock et al. [111] in human SAT. While several studies have reported the presence of transcriptionally diverse populations of vascular cells in human white adipose tissues, how their function may differ among depots and in distinct metabolic and disease states remains a knowledge gap.

### 6.3. Livestock Models

Our recent sc/snRNAseq studies of bovine skeletal muscle and adipose tissue identified both blood (*VWF+*) and lymphatic (*LYVE1*, *MMRN1*) endothelial cells and mural cells (*ACTA2*) [33,116]. Moreover, possibly due to the larger number of genes identified via scRNAseq vs. snRNAseq, clusters corresponding to venular (*AQP1*) and non-venular blood endothelial cells were identified. For the same reason, in beef cattle, we were able to further separate muscle mural cells into pericytes (*PDGFRB*) and arterial (*MYH11*, *CSPG4*) and venous (*MYH11+*, *CSPG4-*) vascular smooth muscle cells. This is similar to what has been previously reported in the adipose tissue of other species, such as human [93,124,132], pig [113], and mouse [93]. In contrast, in dairy cows’ adipose tissue, we were not able to separate pericytes (*NOTCH3*, *PDGFRB*) from smooth muscle cells (*ACTA2*, *MYL9*) using snRNAseq, which together corresponded to nearly 3% of all nuclei across SAT and VAT samples [33].

Besides the common presence of vascular cells among adipose depots, breeds, and species, some differences were also noticed. A larger population of blood endothelial cells was identified in bovine SAT than VAT [33], similar to that observed by Lumeng’s group in human adipose tissue [112]. More abundant vascular cells were found in Wagyu than Brahman skeletal muscle [116]. These differences are highly correlated with physiological differences. The less developed vascular system in VAT vs. SAT likely makes VAT more vulnerable to hypoxia and hypoxia-induced adipocyte apoptosis in VAT. Similarly, the more abundant vascular cells in Wagyu muscle likely contribute to the stronger accumulation of IMAT in Wagyu as the vascular system supports adipogenesis. We found that Wagyu FAPs are more pro-angiogenic and have a stronger communication with endothelial cells through VEGFA-VEGFRs [116]. Future studies could focus on defining the communications between different subtypes of vascular cells and non-vascular cell types, such as ASPC, adipocyte, and immune cell, to understand their regulatory roles in adipose tissue physiology and pathology.

Endothelial cell (*PECAM1*) populations have also been identified in IMAT of pigs, in which animals with higher marbling showing a greater abundance of endothelial cells [117]. Additionally, Wang et al. [113] identified three functionally distinct endothelial cell (*PECAM1*, *VWF*) populations in pig adipose tissue including proliferating endothelial cells (*CENPF*, *CENPE*, *TOP2A*, *TPX2*), immune-active endothelial cells characterized by high expression of complement system genes (*C1QA*, *C1QB*, *C1QC*), cathepsins (*CTSS*, *CTSD*, *CTSB*, *CTSZ*) and cystatins (*CST3*, *CSTB*), as well as a population of endothelial–mesenchymal transitional cells (*ACTA2*, *TAGLN*). In the breast muscle of chickens, a cluster of endothelial cells (*TMSB4X*, *GNG11*, *RHOA*) was identified at 100 days old, but not 5 days old [129]. Similarly to humans, further studies are required to elucidate the vascular cell diversity and function in adipose tissue of animals in distinct productive stages and their potential role on meat quality and disease pathogenesis.

## 7. Mesothelial Cells: A Depot-Specific, Heterogenous Cell Type

The cell type compositions are generally similar among all WAT depots. For instance, studies have suggested that essentially all white adipocytes are derived from *Pdgfra^+^* ASPCs; however, human mesothelial cells also express *PDGF* and *PDGFRA* [133]. In fact, Emont et al. recently demonstrated that *PDGFRA* and *PDGFRB* are expressed not only by human ASPCs, but also by mesothelial cells, mature adipocytes, T cells, macrophages, and endothelial cells [93]. Several studies have reported mesothelial cell-like ASPCs as a VAT-specific cell type, varying in adipogenic capacity depending on their gene profile and location [134]. Mesothelial cells are epithelial cells of mesodermal origin that form a monolayer (mesothelium) lining the visceral serosa [19]. The function of mesothelial cells in adipose tissue remains unresolved and most single-cell studies from mice and humans do not differentiate mesothelial cells from other ASPC types.

An elegant lineage-tracing study by Chau et al. [135] using Wt1-CreERT transgenic mice, characterized the ex vivo adipogenic differentiation of mesothelium and revealed a higher percentage of adipocyte differentiation in intra-abdominal fat depots including epididymal (77%) and epicardial WAT (66%), followed by omental WAT (47%) and mesenteric WAT (28%). Interestingly, no labeled adipocytes were observed in subcutaneous WAT or brown adipose tissue. Consistently, it was recently reported that *Wt1* lineage cells specifically contribute to VAT but not SAT development [101]. Our recent lineage-tracing study identified the unique expression of *Tcf21* in all VAT ASPCs [101]. Single-cell RNA sequencing analysis showed that *Wt1* and *Tcf21* have a similar expression pattern in VAT in a mice model; however, *Tcf21* is specifically expressed in *Pdgfra^+^* ASPCs, while *Wt1* is also expressed in *Pdgfra^−^* cells, likely including mesothelial cells [101].

Hepler et al. [19] also revealed a *Pdgfrb^+^* (CD31^−^/CD45^−^) *Ly6c^−^*/*Cd9^+^* mesothelial-cell like subpopulation in VAT of mice, further characterized by the expression of classical mesothelial cell markers, including *Krt18*, *Pkhd11b*, *Upk1b*, *Msln*, *Krt8*, *Krt19* and *Upk3b.* While VAT mesothelial cells lacked adipogenic capacity in vitro, they did not demonstrate any anti-adipogenic effects, as observed with the *Ly6c^+^/Pdgfr^+^* (FIPs), which inhibited ASPC differentiation. A similar population of mesothelial cells was described by Sárvári et al. [95] in their mouse VAT samples, characterized by high expression of typical mesothelial cell markers *Upk3b* and *Msln*, as well as *Gpm6a*.

In human adipose tissue, Emont et al. [93] identified three distinct *MSLN^+^*/*KRT19^+^* mesothelial subpopulations in human VAT (*GYPA^+^*, *GALNT15^+^* and *LOXHD1^+^*) and mouse perigonadal adipose tissue (*Cdh12^+^*, *Fth1^+^*, *Nalcn^+^*). Only two subpopulations were found to be associated with obesity, with a lower proportion of *GYPA^+^* and a higher proportion of *GALNT15^+^* mesothelial cells in subjects with higher BMIs. While Strieder-Barboza’s work did not identify specific populations of mesothelial cells, the authors report a VAT-specific inflammatory-mesothelial-like ASPC (IM-ASPC) population located on the surface lining of AT, consistent with mesothelium [112]. Additionally, the transcriptional profile of these IM-ASPC were similar to the one reported in *MSLN^+^* VP1/VP3 of Vijay et al. [20], reinforcing a mesothelial origin for IM-ASPCs.

In adipose tissue of dairy cows, we observed a striking difference in the frequency of mesothelial cells (ME; *MSLN^+^*), which were mostly absent in abdominal SAT (0.3% of all nuclei) compared to omental VAT (15% of all nuclei) [33]. These results agree with previous snRNAseq and scRNAseq studies in mouse and human models, which indicate a specific expression of mesothelial markers in VAT [20,93]. In our study with bovine adipose tissue, all ME expressed both *MSLN* and *KRT19*, similarly to the recently reported in human VAT [93]. Sub-cluster analysis of mesothelial cells revealed three transcriptional distinct subpopulations characterized by the selective expression of *WT1* and *UPK3B*: ME1 was *WT1^+^UPK3B^-^*, ME2 was *WT1^+^UPK3B^+^*, and ME3 *WT1^-^UPK3B^-^* [33]. Previous studies in mice have reported that *WT1*^+^ ME cells were VAT-specific ASPCs that become adipocytes [135], while others have indicated that *KRT19^+^* ME were not a source of adipocytes in mice [134]. While we did not evaluate the adipogenic capacity of bovine adipose tissue mesothelial cells, ME2 was enriched for genes associated adipogenesis (*CD34*, *IGF2*), inflammation (*C3*, *CFB*, *C1S*, and *CD99*), and fibrosis (*CD9*, *SPARC*, *COL8A1*) [33]. Thus, establishing the differences between distinct mesothelial cell subtypes and ASPCs is key to understanding the role of these cells on adipose tissue function and dysfunction.

## 8. Conclusions and Future Directions towards the Use of Single-Cell Analyses to Address Livestock Health and Production

The extensive research discussed in this review highlights the role of adipose depots not as inert energy storage sites, but as dynamic tissues with considerable functional and cellular heterogeneity. Building upon the findings of prior FACS and lineage tracing investigations, single-cell and single-nuclei sequencing techniques have been essential for identifying and characterizing adipose tissue cell types, as well as for generating hypotheses linking depot-specific transcriptional and functional differences. While most of the adipose tissue research is performed in the context of human health and disease, there is significant potential for further investigations in livestock with implications for animal health and production. With the increasing application of single-cell omics technologies in the field of adipose tissue biology, we expect more exciting research exploring adipose tissue complexity and identifying species-, breed-, depot-, and age-specific characteristics of adipose tissue.

There is great potential for the application of single-cell technologies in bovine systems, particularly in the context of animal health and productivity. For beef cattle, increased deposition of IMAT (marbling) is highly desirable to improve meat quality. However, since the development and accumulation of IMAT is usually posterior to those of SAT and VAT, promoting IMAT formation through nutritional management is usually accompanied by even greater increases in SAT and VAT, which have minimal economic value. Significant efforts have been made to identify differentially expressed genes among depots, which can be targeted for depot-specific manipulation of fat accumulation. However, due to differences in the complex cell-type composition among depots, it is very difficult to select effective genes using the conventional bulk sequencing techniques and design efficient strategies. Through single-cell analyses, differences in cell types directly contributing to adipogenesis, such as FAPs and adipocytes, can be compared precisely among depots to identify possible genes responsible for differential fat accumulation among depots. Moreover, owing to the high resolution of single-cell sequencing, other unique and common cell types indirectly regulating adipogenesis, such as mesothelial cells, vascular cells, and immune cells may also be subjected to comparative analyses to identify additional contributing factors. These studies may eventually lead to the development of appropriate strategies that specifically increase IMAT formation, while limiting excessive SAT and VAT deposition [136,137]. Moreover, differentially expressed genes identified through comparative studies of different individuals and breeds may serve as markers in marker-assisted breeding, which may facilitate the selection of animals with desired traits.

In dairy cows, the transition between pre- and postpartum remains a critical period that leaves cows vulnerable to a host of diseases as well as metabolic dysfunction [138]. Characterizing this transition period are dramatic changes in lipid and energy metabolism, forcing cows into a state of enhanced lipolysis and limited lipogenesis [139]. While these shifts in metabolic function are necessary to accommodate the sudden demand for milk production, cows that are unable to adequately adapt to these conditions are susceptible to developing metabolic diseases such as ketosis, as well as experience reduced milk production and compromised reproductive performance [140]. Hence, a better understanding of the factors that drive these metabolic changes and maladaptation at the cellular level remains necessary. In line with significant research endeavors into cancer biology and tumor heterogeneity in humans, single-cell applications in dairy cows will provide insight into the types and abundance of cells, as well as specific target genes in different depots and in different production stages or disease conditions. This will aid in the identification of new biomarkers and the development of targeted therapeutic strategies to prevent metabolic dysfunction and disease in dairy cows.

Other than the more widely used sc/snRNAseq technique, other sc/sn omics techniques can be additional powerful tools in the study of cattle and other livestock species. Even though many of these techniques are becoming increasingly popular in the biomedical field, their utilization in livestock research is still sparse. One such technique that has been commercialized is the sc/snATACseq. With the sc/snRNAseq+ATACseq multi-omics strategy, researchers will have the opportunity to study the epigenetic regulation of adipogenesis and other cell activities in cattle and identify key transcription factors regulating these processes.

## Figures and Tables

**Figure 1 biology-12-01289-f001:**
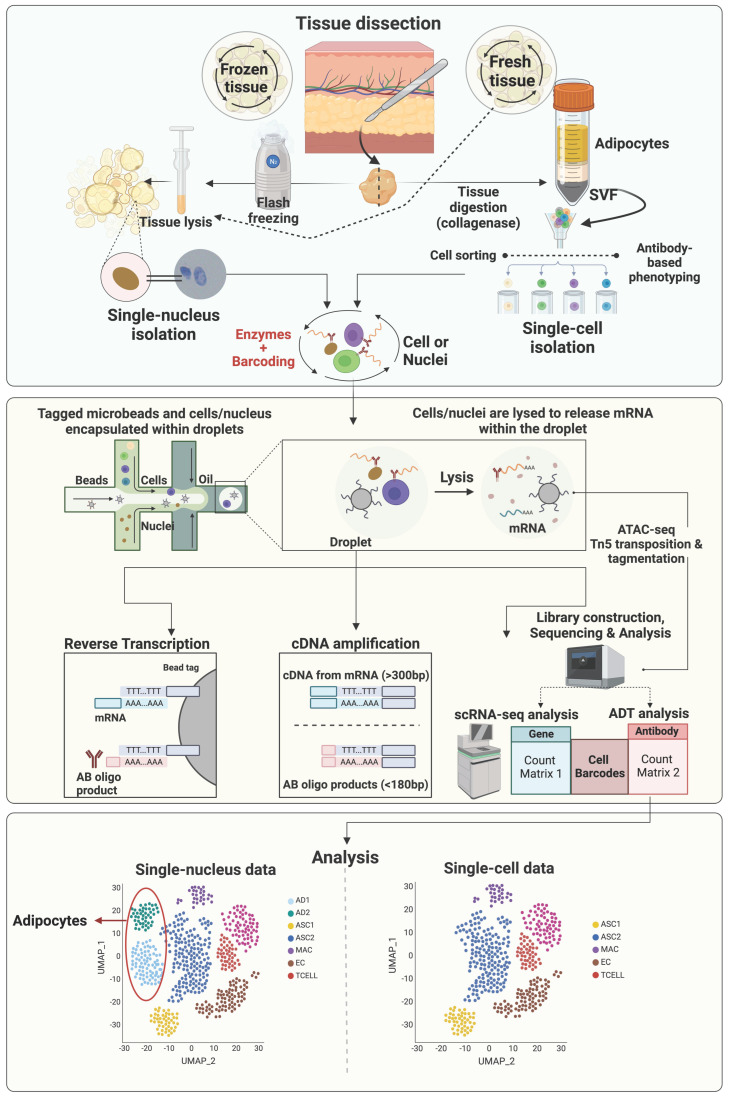
General workflow of single-cell and single-nucleus RNA sequencing. This figure was prepared in Biorender and is licensed to be published (Agreement number: LA25UR522V).

**Figure 2 biology-12-01289-f002:**
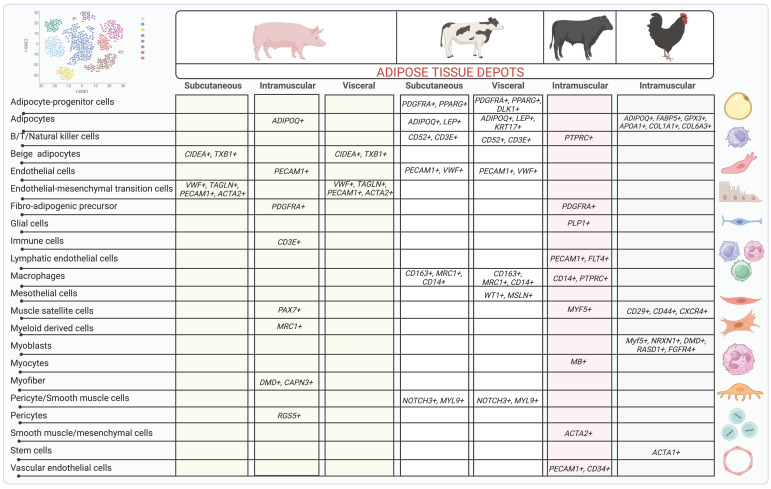
Genetic markers for cell types recently identified in adipose tissue depots of pigs, dairy cattle, beef cattle, and chicken. This figure was prepared in Biorender and licensed to be published (Agreement number: ZL25UR6E87).

**Table 1 biology-12-01289-t001:** Summarized results from recent studies evaluating progenitor cell populations in white adipose tissue depots in different mammalian models.

Publication	Adipogenic or Fibro- Adipogenic Progenitors	Mesothelial/Mesenchymal/Interstitial Progenitors	Other Progenitors	Model
Burl et al., 2018 [105]	General ASPC Classification: *Lin-*/*Pdgfrb+*/*Ly6a+*			Mouse SAT and VAT
	Proliferating ASPC ^1^: *Pdgfra+*/*Cdca8+*			
	Early-differentiating ASPC: *Cebpa+*/*Plin1+*			
	Differentiating ASPC: *Cebpa+*/*Adig+*/*Plin1+*/*Scd1+*			
Schwalie et al., 2018 [106]	General ASPC Classification: CD31-/CD45-/TER119-/CD29+/CD34+/SCA1+			Mouse SAT
	Adipogenic ASPC: *Adam12+*/*Aoc3+*/*Peg3+*/*Fabp4+*	Stem cells: *Creb5+*/*Cd55+*/*Il13ra1+*	Anti-adipogenic cells: *Meox2+*/*Abcg1+*/*F3+*	
Hepler et al., 2018 [19]	ASPC: CD45-/CD31-/PDGFRB+/LY6C-/CD9-	MSC ^2^: CD45-/CD31-/PDGFRB+/LY6C-/CD9+	FIP ^3^: CD45-/CD31-/PDGFRB+/LY6C+	Mouse SAT and VAT
Merrick et al., 2019 [21]	Committed preadipocytes (mice): *Icam1+*/*Dlk1+*/*Pparg+*/*Fabp4+*/*Cd36+*	Interstitial progenitors (mice): *Dpp4+*/*Wnt2+*/*Bmp7+*/*Pi16+*	Adipogenesis-regulatory cells (mice): *Clec11a+*/*Fmo2+*/*F3+*	Mouse and Human SAT
	Committed preadipocytes (humans): *PDGFRA+*/*PDGFRB+*/*SCA1+*/*ICAM1+*/*PPARG+*/*GGT5+*	Interstitial progenitors (human): *PDGFRA+*/*PDGFRB+*/*SCA1+*/*DPP4+*/*CD55+*/*WNT2+*		
Cho et al., 2019 [107]	General ASPC Classification: CD45-/CD31-/TER119-/SCA1+			Mouse VAT
	Committed preadipocytes: Igf1+/Col4a1+/Col4a2+/Sult1e1	Stem cells: *Cd55+*/*Cd34+*/*Fbn1+*/*Anxa3+*/*Mfap5+*/*Timp2+*/*Dpp4+*/*Pi16+*		
Vijay et al., 2019 [20]	General ASPC classification: CD45-/CD31-/CD34+/CFD+			Human SAT and VAT
	Preadipocytes: *MGP+*/*APOD+*/*CXCL14+*/*WISP2+*	VAT-specific mesothelial cells: *ITLN1+*/*MSLN+*	Hematopoietic stem cells: *CCL5+*/*CD3E+*/*IL7R+*/*IL32+*	
	Mature adipocyte progenitors: *APOE+*/*FABP4+*/*CEBPB+*/*CD36+*		Fibrotic and ECM-associated cells: *COL3A1+*/*COL6A3+*/*COL1A1+*/*COL6A1+*	
Raajendiran et al., 2019 [108]	General ASPC Classification: CD45-/CD31-/CD29+			Human SAT and VAT
	CD34^high^: *APOD+*/*ICAM1+*	CD34^low^: *ITLN1+*/*PLA2G2A+*/*UPK3B+*	CD34- beige adipocytes: *PRDM16+*/*UCP1+*	
Sárvári et al., 2021 [95]	FAP: *Col1a1+/Pdgfra+/Dcn+*	Mesothelial cells: *Upk3b+/Msln+/Gpm6a+*		Mouse VAT
Hildreth et al., 2021 [109]	ASPC: *PRG4+/DKK1+/PI16+/PDGFRA+* Preadipocytes: *PDGFRA+/CXCL14+/GPC3+*			Human SAT
Shan et al., 2022 [110]	VAT ASPC: CD45-/CD31-/PDGFRB+/LY6C-/CD9-		VAT FIP: CD45-/CD31-/PDGFRB+/LY6C+	Mouse SAT and VAT
	SAT ASPC: CD45-/CD31-/PDGFRB+/DPP4±			
Whytock et al., 2022 [111]	Preadipocytes: *ATXN1+/ZNF423+/CD38+*	General stem cells: *PTPRC-/PECAM1-/CD34+/PDGFRA+/PDGFRB+*		Human SAT
Emont et al. 2022 [93]	SAT ASPC: *CEBPD+*/*SGCZ+*	Multipotent progenitors: *PDGFRA+*/*ALDH1A3+*	Anti-adipogenic cells: *PDGFRA+*/*EPHA3+*	Mouse and Human SAT and VAT
	VAT ASPC: *FGF10+*/*PDE4D+*			
Strieder-Barboza et al., 2022 [112]	ASPC: CD45-/CD31-/TM4SF1- (*LUM+*/*DCN+*/*CFD+*/*APOD+*/*CD142+*/*MFP5+*/*S100A4+*)	Inflammatory mesothelial-like ASPC: CD45-/CD31-/TM4SF1+ (*PLA2G2A+*/*SLPI+*/*ITLN1+*/*TIMP1+*/*KRT8+*/*MSLN+*)	Pro-fibrotic precursors: *COL1A1+*/*COL6A1+*/*FN1+*/*LOX+*/*LUM+*	Human VAT
Michelotti et al., 2022 [33]	Committed preadipocytes: *PPARG+*/*SLC1A3+*/*LIPE+*/*GPAM+*/*LMO4+*	VAT-specific mesothelial cells: *MSLN+*/*KRT19+*/*WT1+*/*UPK3B+*		Bovine SAT and VAT
	FAP: *PDGFRA+*/*FBN1+*/*FN1+*/*LAMA2+*/*COL14A1+*/*MFAP5+*			
Wang et al., 2022 [113]		Endothelial–Mesenchymal transition cells: *VWF+*/*TAGLN+*		Porcine SAT and VAT
Garritson et al., 2023 [114]	General FAP ^4^ Classification: *PDGFRA+*			Mouse and Human VAT
	Uncommitted progenitors: *PI16+/SEMA3C+/OSR2+*		Anti-adipogenic cells: *THY1+/CTHRC1+*	
	Committed preadipocytes: *ICAM1+/CEBPB+*		Pro-fibrotic precursors: *VCAN+/MFAP5+/HTRA3+*	
Fitzgerald et al., 2023 [115]	General FAP Classification: *PDGFRA+*/*CD34+*/*DCN+*/*NCAM1-*			Human IMAT
	Adipogenic FAP: *MME+*/*PTGDS+*/*CXCL14+*/*SMOC2+*		Pro-fibrotic precursors: *CD55+*/*TNXB+*/*MFAP5+*/*PCOLCE2+*/*FBN1+*/*PRG4+*	
	Transitional FAP: *GPC3+*/*SFRP2+*			
Liu et al., 2023 [101]	ASPC: *Tcf21+*/*Pdgfra+*/*Icam1+*/*Dpp4+*			Mouse VAT
Wang et al., 2023a [116]	General FAP Classification: *PDGFRA+*			Bovine IMAT
	Uncommitted FAP: *IGFBP5+*/*NUPR1+*		Fibrogenic FAP: *POSTN+*/*TGFBR3+*/*COL1A1+*	
	Transitional FAP: *DLK1+*/*ACKR2+*/*PHLDA2+*			
	Adipogenic FAP: *CFD+*/*BTG1+*/*BMP4+*/*COL4A1+*			
Wang et al., 2023b [117]	FAP: *PDGFRA+*			Porcine IMAT

^1^ ASPC = adipocyte/stem progenitor cells; ^2^ FIP = fibro-inflammatory progenitors; ^3^ MSC = mesenchymal stem cells; ^4^ FAP = fibro-adipogenic precursors.

## Data Availability

No unpublished data presented in this review.

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
