# Peer review of "White Adipose Tissue Heterogeneity in the Single-Cell Era: From Mice and Humans to Cattle"

_biology, 2023, doi:10.3390/biology12101289_

Round 1

Reviewer 1 Report

Streider-Barboza and colleagues have put together a comprehensive review aimed at providing a deeper description of the various adipose depots in rodent, livestock and humans with an emphasis on further deciphering the unique cellular compositions of those depots. Overall, the review is very well-written and provides evaluations of emerging technology and data in the field of single cells and single nuclei with a particularly unique emphasis on livestock research. The inclusion of the livestock research with the human research is compelling, and the authors provide an extensive list of references for the readers to explore in more detail while also providing solid summations of the literature where appropriate. In my opinion, the true strength of this review is the in-depth historical perspective of the emergence of the single cell technology, in particular as it is applied to the complex tissue of adipose, as well as including the livestock data which more often than not is somewhat ignored or missed in the field of adipose biology. In sum, I think this is a very nice and comprehensive review for this field. I would ask the authors to expand a bit more in their Conclusions on where they think this field is going and how these data referenced herein have shaped the current view of the field.

My single major critique I have is that the review is hard to follow at times as it winds through the different depot descriptions, as well as the descriptions of adipokines then progenitor cells then on to the single cell work. In my opinion, the sections on adipokines could be completely removed (Section 2.2 on Endocrine function) and the authors could expand the descriptions of the other cell types in the models using the single cell/single nuclei data. I also think the Section 2.1 could considerably be reduced in terms of the descriptions on innervation and blood supply. As they’re described now, it’s not nearly as in-depth as the rest of the review, so seems superfluous by comparison. The authors do a nice job of comparing and contrasting progenitor cells and mature adipocytes among the different models and depots from the literature, but could perhaps also discuss some of the other non-adipocyte cells such as the repertoire of immune cells or endothelial cells, etc.

Here are some minor comments:

1.     The section on mature adipocytes might also benefit from having a similar structure as the progenitor cells (ie broken out by models).

2.     Some key references are missing in the single cell section. There are a number of single cell and single nuclei papers in human adipose tissue missing from groups such as Ryden, Spalding, Sparks, etc.

3.     There have been some recent reviews on IMAT in humans and across species that are also not referenced – Goodpaster, Bergman, Sparks, etc

4.     Line 187 – “and” is missing from the sentence between “mouse” and “in humans”.

Reviewer 2 Report

This review article describes the state of knowledge on several topics relating to white adipose tissue physiology, with the purpose of presenting what is currently known about cattle. The authors describe white adipose tissue depots, adipokines secreted from white adipose tissues, as well as information about what is currently known about WAT depot-specific cell composition and function. The authors conclude that their findings support a need to further explore omics techniques to garner a greater understanding of bovine adipose tissue physiology for the purpose of improving livestock. Several significant weaknesses diminish enthusiasm for publication of this work in its current form. 

Major Comments:

1)    Focus of manuscript- This manuscript covers a lot of different topics, making it difficult to distinguish what the true focus of the article is. After reviewing this manuscript in its entirety, the main focus of the article appears to be the omics approaches sections. The section earlier in the review focused on adipokines does not seem very relevant or connected to the latter sections. This reviewer suggests that the authors deeply consider the main focus of their article and remove any relevant topics or information that does not directly connect to it. This may require significant restructuring of the article.  

2)    Abstract- The abstract does not accurately represent the topics that are covered in the manuscript. Notably, ¼ of the abstract is information about recent publications from the lab, rather than topics discussed in the literature review.

3)    Parallel structure/ logic and flow-  Based on this reviewer’s understanding, the novelty of this review paper is that it compares present findings with the present state of knowledge in cattle. Throughout the manuscript, findings are described in one organism, and then not mentioned or described in the same level of detail in cattle, or vice versa. It is important to describe results in a parallel structure- for example, if a specific concentration of an adipokine is mentioned in humans, it should also be mentioned- or acknowledged that the average is unknown- in mice. This creates a structure to the information that is understandable to the reader.

4)    Clarification about species each study is carried out in.  in general, there is a lot of back-and-forth between studies done in different organisms. For the understanding of the reader, it is important to make it clear what organism each finding is from. Example: line 345. The prior sentence describes an in vitro study in bovine cells, while the cited study is in humans, but it is not mentioned. Please make sure that this is clearly stated throughout the manuscript, not just in this specific example.

5)    Clarification- Line 70 and elsewhere- that this review describes white adipose tissue function. Example: Line 70- “the two main types of adipose tissue depots in animals” – which is untrue because brown adipose tissue depots also exist. Please clarify throughout the manuscript.

6)    Figures- It is illegal to use image software without citing the source. Please be sure to give credit to Biorender for images. Biorender provides information about how to do this under copyright law. Please cite biorender and/or other imaging softwares in each figure legend, and acknowledgements if it is required by the journal.

Minor Comments: Please note that these comments are no longer relevant should the authors choose to remove sections of the manuscript to improve upon the focus of the text.  

1)    Line 187: “in mouse in humans” – this reviewer is confused about which is correct; please revise.

2)    Line 199- the authors cite specific values for leptin levels in dairy cows but do not provide specifics in humans for comparison. It would be helpful to provide the same level of detail in comparisons.

3)    Line 215- the term ‘adipokine’ has already been defined earlier in the text, and saying that adiponectin is produced and secreted by adipocytes is redundant.

4)    Line 234- stated above that AdipoR1 and AdipoR2 are expressed in skeletal muscle and liver, respectively. Are these also expressed in adipose tissue? Please clarify and cite evidence about where they are expressed in dairy cows.

5)    Line 251. Please clarify cell specificity for NFKB suppression by adiponectin, and include information about species that these experiments are conducted in.

6)     conclusion is not really supported by the information cited. Please revise the statement. Is this statement made in reference #88?

7)    Line 254- is this statement supported by the cited information? Is this not true in other organisms?

8)     Is there parallel information with regard to lactation in either mice or humans or adipose depot specificity (line 285-6, 290) for comparison?

9)     the authors compare the pathologies associated with human IMAT and then provide a comparison that it provides flavory effects to meat in cows. Are there any pathologies associated with IMAT, to provide a parallel? These two things are not comparable, but it Is worth presenting that in cows it is thought about from a different perspective.

10)  in general, there is a lot of back and forth between studies done in different organisms. Wherever appropriate, please make sure it is obvious whether the study is done in humans, rodents, etc. Here, please specify that these findings are from humans.

11) Line 351- do the authors mean gene expression or protein levels? How was this measured? Usually gene expression is in Sentence case, with only the first letter capitalized, but this depends what organism it is in. Please double check and provide clarification  on this matter.

12) Line 382- please mention the markers used and the cell type for each type of cell identified.

13) Fig 1. Some of the text is too small to see. Please make figure text larger so that it is legible in print.

14) Table 1. For tables that span multiple pages, it is helpful to list the top row at the top of each page. Terms used in the table should be defined in the table legend.

15) Figure 2. text is completely illegible even blown up on a computer screen.

English is overall satisfactory; the manuscript is well written.

Round 2

Reviewer 2 Report

The authors comprehensively restructured their manuscript and addressed all reviewer comments. There are no further comments for review.